# Volatilome Analysis and Evolution in the Headspace of Packed Refrigerated Fish

**DOI:** 10.3390/foods12142657

**Published:** 2023-07-10

**Authors:** Doriane Martin, Catherine Joly, Coralie Dupas-Farrugia, Isabelle Adt, Nadia Oulahal, Pascal Degraeve

**Affiliations:** BioDyMIA Research Unit, Université de Lyon, Université Claude Bernard Lyon 1, ISARA Lyon, 155 Rue Henri de Boissieu, F-01000 Bourg en Bresse, France; doriane.martin@univ-lyon1.fr (D.M.); catherine.joly@univ-lyon1.fr (C.J.); coralie.dupas-farrugia@univ-lyon1.fr (C.D.-F.); isabelle.adt@univ-lyon1.fr (I.A.); nadia.oulahal@univ-lyon1.fr (N.O.)

**Keywords:** fish preservation, fish packaging, monitoring of fish quality, volatilome analysis

## Abstract

Fresh fish is a perishable food in which chemical (namely oxidation) and microbiological degradation result in undesirable odor. Non-processed fish (i.e., raw fish) is increasingly commercialized in packaging systems which are convenient for its retailing and/or which can promote an extension of its shelf-life. Compared to fish sent to its retail unpackaged, fish packaging results in a modification of the gaseous composition of the atmosphere surrounding it. These modifications of atmosphere composition may affect both chemical and microbiological degradation pathways of fish constituents and thereby the volatile organic compounds produced. In addition to monitoring Total Volatile Basic Nitrogen (TVB-N), which is a common indicator to estimate non-processed fish freshness, analytical techniques such as gas chromatography coupled to mass spectrometry or techniques referred to as “electronic nose” allow either the identification of the entire set of these volatile compounds (the volatilome) and/or to selectively monitor some of them, respectively. Interestingly, monitoring these volatile organic compounds along fish storage might allow the identification of early-stage markers of fish alteration. In this context, to provide relevant information for the identification of volatile markers of non-processed packaged fish quality evolution during its storage, the following items have been successively reviewed: (1) inner atmosphere gaseous composition and evolution as a function of fish packaging systems; (2) fish constituents degradation pathways and analytical methods to monitor fish degradation with a focus on volatilome analysis; and (3) the effect of different factors affecting fish preservation (temperature, inner atmosphere composition, application of hurdle technology) on volatilome composition.

## 1. Introduction

Total fisheries and aquaculture production reached a record 214 million tons in 2020, which corresponds to a significant increase in the world’s consumption of aquatic foods [1], despite their rather high global warming potential or carbon footprint [2].

To avoid unnecessary waste, it is needed to prevent fish from premature spoilage during fish processing from the time of capture of fish to its sale. Several and varied technological routes can be listed such as chilling, freezing, or the ancient storage processes such as salting, drying, smoking, or the rather new treatment of HHPP (HHPP: High Hydrostatic Pressure Processing), or the addition of chemical preservatives or new alternative natural ones [3]. Moreover, appropriate multilayer packaging films with precise operational specifications are now available for AP (Air Packaging), VP (Vacuum Packaging), or MAP (Modified Atmosphere Packaging [4]) and are supposed to contribute to the preservation of fish, but the effects of anaerobic or carbon dioxide conditions are still being investigated [5].

Indeed, fresh fish is a highly perishable food with a short shelf life (even under refrigeration) compared to other animal protein foods, such as meat and eggs, because it has neutral pH, high moisture content, and larger quantities of non-protein nitrogen molecules, all of which provide ideal conditions for microbial and biochemical spoilage [5].

Negative changes in fish odor, flavor, texture, and possible severe food safety concerns are associated with three basic mechanisms: enzymatic autolysis, oxidation, and microbial degradation [5,6,7,8,9]. Quality changes and the shelf life of chilled fish are thus well documented and routinely achieved through time consuming analyses more often at the end of the fish shelf life: (i) sensory methods (discriminative tests and descriptive tests); (ii) biochemical and chemical methods (Total Volatile Basic Nitrogen (TVB-N), ammonia, TriMethylAmine (TMA), DimethylAmine (DMA), biogenic amines, nucleotide catabolites, and ethanol assays, measurements of oxidative rancidity; (iii) physical methods (pH, texture analysis); and (iv) microbiological methods (total viable counts, spoilage bacteria and reactions, pathogenic bacteria detection and/or enumeration).

There is an abundant literature investigating the aging of the tryptic (fish product + packaging + temperature): these studies were carried out with multi-criteria characterizations under given environmental conditions, which could be assigned as “case study”, in most cases in order to determine the shelf lives upon quality and standardized safety criteria. For example, fish can be stored at a constant temperature (from chilled to super chilled fish [10,11]) or at variable temperatures according to the shelf life evaluation protocol monitored by industrial plants [12], ionized and then inoculated by identified spoilage microorganisms [13], etc.

More recently, the analysis of the composition of the volatilome [14,15,16,17,18] has also become significant. After a first application in the field of food safety (adulteration), the volatile organic compounds (VOCs) in food became important giving knowledge about the quality of foods and their relationship to consumers’ choices. However, food aroma is a mixture of varied molecules (alcohols, aldehydes, acids, esters, terpenes, etc.) for which isolation, identification, and quantification can be challenging [19].

Additionally, fish volatile compounds evolution can be investigated in packed fish storage. In this case, the packaging nature (in most cases polymeric multilayer materials when authors used commercial packaging solutions) and conditioning techniques [20] may also vary largely from air to vacuum packaging with or without modified atmosphere packaging, with different initial atmosphere compositions, with high percentage of CO_2_ to reduce microbial growth of common aerobic bacteria, while oxygen is required to inhibit the growth of *Clostridium botulinum* type E [4,21,22].

In this context, this review aims to provide insights into:(i)The packaging systems used to generate typical headspace gas composition evolution during packed fish shelf life and containment of VOCs according to packaging materials barrier properties.(ii)The fish spoilage mechanisms generating VOCs with emphasis on their chemical structure, properties, and possible origin. VOCs are mainly generated via enzymatic reactions, lipid oxidation, and microbial actions.(iii)A comparison with the traditional methods to assess fish freshness (e.g., TVB-N or TMA assay), which will be detailed thereafter.(iv)The analytical methods allowing the isolation, identification, and assay of VOCs present in the headspace of a packaging, listing for each method their advantages and limits.

This review aims thus to draw up a state of the art on the research on VOCs evolution during packaged fresh fish storage to identify other relevant fish spoilage markers, that namely could detect fish spoilage at an earlier stage than following commonly used TVB-N or TMA assays.

## 2. Headspace Atmosphere Composition as a Function of Packaging

The headspace atmosphere composition, namely O_2_ and CO_2_ percentages, is a rather simple analysis implemented to study the fresh fish spoilage when packed under air (AP), vacuum (VP) or modified atmosphere (MAP) [23]. MAP of fish generally relies upon an increased CO_2_ concentration, from 20 to 80% [4], this gas being reported as extending the lag and generation times of aerobic bacteria, yeasts, and molds, thereby reducing their metabolic activity and growth [24]. Indeed, it is well-known that VP and MAP extend the shelf life of refrigerated fish (generally stored between 0 °C and 4 °C) from 2 days to 2 weeks [25,26] or even longer if stored at super-chilling temperatures (−2 °C) [10]. The headspace atmosphere composition evolves more or less rapidly over time and in a predictable but not very well controlled manner. It is not yet possible as for fruit or vegetables [27] to control or predict the inner atmosphere composition versus time or to reach an equilibrated (constant) carbon dioxide and/or oxygen concentration: only the initial atmosphere composition, voluntarily added, is well-known. Figure 1 shows the schematic evolution of the gaseous composition of the headspace as a function of time starting from the initial composition for AP, VP, or MAP. As expected, for all conditions, the oxygen concentration decreases as the CO_2_ concentration increases for packaging with gas barrier properties. This behavior is well-known and has already been reported to be due to the establishment of correlated complex phenomena, such as physico-chemical, biochemical, and biological alterations of the fish matrix, rich in water, sensitive to oxidation, to light, etc., and by the metabolism of the endogenous microbial flora. Indeed, aerobic microbial respiration (consumption of O_2_, generation of CO_2_) is often presented as a determining factor of the gaseous composition evolution for short storage times [28]. For example, for VP, the initial composition is one of the entrapped residual airs but it turns rapidly to anaerobic conditions [29] without possible extended oxidation as there is obviously quasi no headspace.

In fact, these evolutions are not easy to model or predict, since they depend on numerous parameters, such as the headspace to food volume ratio, the packaging intrinsic gaseous permeability coefficients for O_2_ and CO_2_, O_2_ and CO_2_ solubilities in multiphasic food matrices [30], the temperature, the geometry of the packaging (i.e., thicknesses, exchange surfaces), the initial composition of the atmosphere, as well as the instantaneous partial pressures differences between external environment (air) and inside the packaging, and the change of spoilage pattern of fish product from one dominated by aerobic bacteria to one dominated by slower-growing facultative anaerobes [31]. Indeed, as the CO_2_ concentration in the headspace increases, the degree to which aerobic spoilage microorganisms are inhibited increases [30,32].

In the literature, the analysis of the headspace composition of fresh packed fish is mainly studied with destructive or non-destructive technologies [25,26,32,33,34], rarely under vacuum for obvious technical reasons related to the lack of headspace volume to be analyzed. The packages selected are generally sealable commercial ones. Air packaging limit is the absence of prevention of any microbial contamination during storage and will thus not be described further here. The packaging materials used for VP and MAP are selected in order to meet three major challenges: (i) strongly limit the inflow of oxygen, (ii) retain a modified atmosphere, and (iii) limit the release of odorant VOCs. For this purpose, multilayer and/or thick materials (trays) are selected [35] on the basis of their intrinsic barrier properties to gases (mainly O_2_, CO_2_, and water vapor).

Fishes or fish fillets are usually packed (MAP and VP) into packaging systems made of a sealed top and bottom films (or trays) previously flushed with selected modified atmosphere (generally a CO_2_ enriched atmosphere) or under moderate vacuum by evacuating the air for a top film having the shape of the food product. As already stated, the packaging acts as a gas barrier packaging limiting gas exchanges (and volatile compounds loss). In general, considering the gas transport properties at steady state (permeability coefficients), the tray is thick enough to act as a nearly total barrier (over 9–12 days), whereas the upper film requires a high degree of technicity (generally 3 to 5 different plastic layers) to reach the same barrier properties, while also ensuring other technical functions such as peelability, mechanical strength, sealability, etc. Meanwhile, the barrier properties achieved by multilayer materials were generally designed from the only intrinsic O_2_ permeability of the inner thin layer made of EVOH (copolymer of Ethylene vinyl alcohol) or PA (PolyAmide) protected from water hydration by two polyolefin external layers such as polyethylene, which play the additional role of sealable water barrier. PA is sometimes selected for its excellent resistance to mechanical strains and is sufficient enough for shorter shelf life or thicker packaging. Until now, EVOHs are considered to be the best oxygen barrier on the market (acting as a solubility barrier) and by extension or habits of the professionals, as a barrier to non-polar or semi-polar VOCs [36]. For example, typical packaging compositions for VP and MAP are stated in Table 1. The given gas transport properties are sometimes only indications of the barrier level, sometimes with confusions between (for example for oxygen) the oxygen flux (cm^3^.m^−2^.day^−1^), the oxygen transmission rate (OTR, cm^3^.m^−2^.day^−1^.atm^−1^), or oxygen permeability coefficient (OPC, cm^3^.cm.m^−2^.day^−1^.atm^−1^), the units of each quantity allowing the precise identification [37]. It should be noted, however, that the permeability values given in the literature are most often taken from technical data sheets. They are often overestimated because they are determined according to the current standards of measurement of OTR and WVTR (water vapor transmission rate) (i.e., at 23 °C), sometimes far from the real conditions of storage of refrigerated food products. As already presented in Figure 1, the headspace of packed fish evolves to a gas composition rich in CO_2_ and anaerobic conditions before the use-by date, since packaging with high gas barrier properties currently used for fresh fish preservation for a storage time of about 8–12 days makes O_2_ and CO_2_ exchanges negligible especially for oxygen.

## 3. Fishes Alteration Criteria

Due to their composition and their physico-chemical characteristics, fishes are highly perishable [7,38]. Major fish spoilage can be attributed to microbial sources. However, degradation processes imply various (bio)chemical modifications which lead to unacceptable organoleptic changes (color, odor, or texture) for consumers [10,39], particularly at the end of their shelf life. Thus, various non-microbial parameters can be used as quality indicators. Among these, TVB-N (resulting from proteins degradation by decarboxylases), TMAO-N, and TMA-N (the two major volatile bases), TBARs (mainly used indicator for lipid oxidation), pH, and sensory evaluation are considered as the main quality indicators for fishes [38,39]. Fish spoilage is rapid: under tropical conditions and without any storage precautions, spoilage begins within 12 h after the fishes were caught [18].

The production of metabolites during fish spoilage is dependent on the packaging conditions during shelf-life. For example, sulfur compounds associated to putrefaction odor (such as H_2_S) are characteristic of storage in the presence of oxygen, while TMA is increased in MAP packaging [35] as well as under the conventional storage in air. Thereafter, we propose an overview of the main fish degradation indicators during conventional storage (without any MAP or vacuum packaging systems, such as storage in ice).

### 3.1. Quality Indicators of Fish Protein Degradation

Sensorial defects associated to fish spoilage are unpleasant odor (marine seaweed odor is associated to fresh fish, while putrid or ammoniacal odors are correlated to a high fish deterioration), color alteration, and decreasing cohesiveness of muscles during storage (thus, the firmer the texture, the more the fish can be considered fresh) [39,40]. Concerning fish muscle deterioration, although some studies suggested that these can be correlated to myobrillar protein alteration, lipid deposition in muscles also seems to be an important parameter [39]. In the same way, some studies have shown that water holding capacity (WHC) decrease can be correlated to poorer muscle characteristics. It has also been reported that an increase of TVB-N of fish leads to discoloration of muscles [39], especially on the back section of fishes. pH can also be considered as a good indicator of fish quality (a pH below 6.8 to 7 is considered as normal for fresh fish; pH higher than 7 is considered as a sign of freshness) [39].

As stated in the previous section, during fish storage, nitrogen-based molecules (mainly proteins) are deteriorating in volatile basic compounds. TVB-N value is the result of fish proteins and TMAO degradation in ammonia, DMA, and TMA. Among these, TMA is a biogenic amine generally associated to unpleasant fish odor (and considered as non-fresh fish) generated by the degradation of TMAO of fish muscle by bacteria and also by endogenous enzymes [18,35]. In fresh fish, TMA level is null or very low. TVB-N and TMA values (which are running in parallel) are commonly used as fish deterioration indicators. However, some studies revealed that the use of TMA and/or TVB-N as indicators of fish freshness is not always relevant [18]. It has been shown that TMA is not a great indicator of degradation during early stage of spoilage for certain fish species. In the same way, although TVB-N is considered a better marker than TMA, results can only be considered after 10 days of storage. This parameter is thus only relevant for measurements of advanced stages of spoilage [41]. Furthermore, it has been established that levels corresponding to spoiled fishes are species-dependent; however, upper limits at 30–35 mg per 100 g and 10–15 mg per 100 g for TVB-N and TMA-N, respectively, can be considered as good indicators for fish spoilage characterization [7,38]. Concerning regulations, maximum values are determined by species. For example, the European Commission (European Union law 95/149/EC, 1995) has fixed a TVB-N upper limit at 25 mg/100 g for *Sebastes* and 35 mg/100 g for *Salmo salar*.

TMA is not the only biogenic amine which can be found in spoiled fish, we can notably cite histamine, putrescine, and cadaverine. All these molecules are the result of amino acids decarboxylation or amination and trans-amination of aldehydes and ketones [38]. Due to the toxicity of histamine, regulations are stricter and a maximum limit has been established at 5 mg/100 g by FDA and at a maximum 20 mg/100 g by European Union (with the mean sample limit at 10 mg/100g) [18] (Official Journal L 268, 24/09/1991 P. 0015-0034, Council Directive 91/493/EEC of 22 July 1991 laying down the health conditions for the production and the placing on the market of fishery products).

### 3.2. Quality Indicators Markers of Fish Oxidation

#### 3.2.1. Analytical Methods

Fish lipid oxidation is a marker of fish quality. As expected, this indicator is more relevant for oily fishes than for whitefish. Indeed, oily fishes’ fillets may contain up to 30% oil/fat. They include species such as mackerel, anchovies, herring, salmon, sardine, cod, swordfish, or trout. Because of the richness in polyunsaturated fatty acids of these fish tissues, this quality indicator is also very important for fish nutritional value assessment, as they are prone to oxidation, especially in processed fish fillets. For example, the catalysis of hematin compounds (hemoglobin, myoglobin, and cytochrome) produces hydroperoxides (non-enzymatic oxidation), as a part of the process of lipid oxidation that also occurs in fish muscle when hemoglobin is deoxygenated or oxidized [18].

Fish lipid oxidation is classically studied using the Thiobarbituric Acid Reactive Substances (TBARs) assay (expressed in mg malondialdehyde (MDA) per kg of lipid) and the Free Fatty Acid (FFA) value (expressed in milliequivalents (mEq) peroxide per kg of lipid or percentage of oleic acid): an increase of these parameters is often associated with fish rancidity [18]. Basically, TBARs assay consists in the spectrophotometric determination of the pink, fluorescent MDA-thiobarbituric acid (MDA-TBA) complex produced after reaction with 2-thiobarbituric acid (TBA) at low pH and high temperature [42]. FFAs are obtained as a side product of triglyceride molecules breakdown and are considered an indicator of lipolytic enzymes (triacyl lipase, phospholipase) presence [18]. FFA may be evaluated by titration with a low concentrated potassium hydroxide solution after fat extraction, colorimetric assays, or even thin-layer chromatography [43]. Some more recent methods feature mass spectrometry: FFA increase and triacyl glycerol increase were monitored during salmon storage for 10 days using these techniques [18].

The main drawback of these methods is that they evaluate the latest steps of the oxidation process. Other methods such as peroxide value, conjugated dienes, p-anisidine value, or Totox (Total Oxidation) value provide information on intermediate steps but are more difficult to implement because the influence of the sample extraction steps is worsened by the transiency of the measured compounds. Indeed, lipid oxidation is mainly a chain reaction, divided into three main stages: initiation, propagation, and termination. During initiation, free radicals are slowly formed from fatty acids under the effect of catalysts such as heat or metal ions. Propagation is the second step, in which free radicals react quickly with oxygen and lead to peroxyl radical, then to hydroperoxide formation, when reacting with other lipid molecules. It is worth noting that fish processing may also lead to different hydroperoxide formation, under lipoxygenase action liberated during tissue cutting [44]. Finally, termination is the final stage where non-radical products influencing taste, color, flavor, and odor (alcohols, aldehydes, acids, ketones, etc.) are formed [45]. These include MDA, measured with a TBARs assay.

Depending on the food matrix on which it is used, a TBARs assay may often overestimate oxidation, since TBA may also react with other compounds of food samples, such as amino acids, carbohydrates, some pigments, etc., and more generally will react with any carbonyl group. It is also often criticized for promoting auto-oxidation of samples because of the strong acidic, long incubation time and high temperature conditions required. This may be amplified by fish sample preparation: cutting and grinding, increasing tissue exposure to oxygen, enzymes, and light (which causes additional peroxide formation by photo-oxidation), even in the presence of added protective antioxidants [46].

To partially overcome these issues in fish samples, more sensitive assays using MDA specific detection techniques, such as HPLC detection after derivatization of MDA with 2,4-dinitrophenylhidrazine (DNPH) and conversion into pyrazole and hydrazone derivatives, have been developed and adapted for fish TBARS determination [39].

#### 3.2.2. Correlation between Volatilome Composition and Fish Rancidity/Oxidation Indicators

Some studies tried to establish correlations between fish volatilome and rancidity indicators, and Figure 2 shows the three most commonly encountered families of molecules.

Cultured and wild sea bream (*Sparus aurata*) were compared for differences in their volatile components over a 23-day storage period in ice (whole fishes in polystyrene boxes with flaked ice) and the volatilome was investigated using Dynamic Headspace Analysis/Gas Chromatography–Mass Spectrometry (GC-MS). Alcohols such as hexanol, heptanol, and aldehydes (hexanal, heptanal, octanal, and nonanal), also known as flavoring compounds, were cited as mainly coming from oxidation or autoxidation of lipids, or enzymatic reactions in sea bream [47].

Chilled Atlantic horse mackerel (*Trachurus trachurus*) minced muscle volatilome oxidation markers were investigated using solid phase extraction combined with GC-MS and compared with peroxide value and TBARS. 1-octen-3-ol was described as one of the best markers of lipid oxidation; its odor is described as rancid, plant-y, and earthy. 1-octen-3-ol also increased during storage of sea bream fillet under air at 0 °C and 15 °C [48] and during 23 days of storage of sea bream in ice [47]. The occurrence of this compound was associated with the oxidation of polyunsaturated fatty acids (namely arachidonic acid) into short-chain volatile compounds by 12-lipoxygenase from fish tissues [49,50], and is also known as a degradation product of linoleic acid hydroperoxides [47]. Cis-4-heptenal results from oxidation of unsaturated fatty acids [50], while heptanal and nonanal arise from the oxidation of n-6 and n-9 polyunsaturated fatty acids, respectively [49,51].

Changes after 10 days storage at 4 °C in volatile compounds were monitored in cod (*Gadus morhua*), whiting (*Merlangius merlangus*), and mackerel (*Scomber scombrus*) and related to spoilage using headspace/mass spectrometric (HS/MS) analysis and solid-phase microextraction/gas chromatographic/mass spectrometric (SPME/GC/MS) analysis. Trans-2,cis-6-nonadienal and trans-2-octenal were associated with the action of 12-lipoxygenase on eicosapentaenoic acid [49]. Trans-2-octenal was also described as responsible for part of the fishy and rancid off-flavors in fresh mayonnaises prepared with fish oil [52]. Other compounds associated to fish rancidity were 1-penten-3-ol, 2,3-pentanedione, propanal, and hexanal [50].

### 3.3. Microbial Spoilage of Fish

Microbial growth and activity are the main factors limiting the shelf life of seafood. Indeed, fish spoilage generally results from off-odors and off-flavors resulting from bacterial metabolism [53]. Therefore, a total viable psychrotrophic count of 10^6^ cfu (colony forming units)/g is generally the upper limit accepted for fresh fish, while spoilage is detectable when the psychrotrophic count exceeds 10^7^–10^8^ cfu/g [54]. Indeed, only a limited fraction of bacteria is responsible for the generation of these off-odors and -flavors. Therefore, the total number of bacteria does not always correlate with fish spoilage. Fish spoilage is better correlated with the growth of specific spoilage organisms. However, organisms producing off-odors and off-flavors are not always identified. Some specific spoilage organisms are not able to produce off-odors and off-flavors in sterile fish, but only following the action of other microorganisms on this substrate. This makes their identification more complex. As reviewed by Gram and Huss [53], specific spoilage organisms not only depend on fish origin (freshwater or marine fish from tropical waters to arctic waters) but also on their refrigerated storage conditions: storage on ice in air, VP, or MAP. Stamatis and Arkoudelos [55] monitored the quality of sardine fillets for 15 days storage at 3 °C in air, under vacuum, or in a 50% CO_2_/50% N_2_ initial atmosphere. The shelf life of sardine fillets estimated based on sensory evaluation was 5, 7, and 9 days for air packaged, VP, and MAP samples, respectively. Sardine bacteria by decreasing order of occurrence were *Shewanella putrefaciens*, pseudomonads, *Brochothrix thermosphacta*, lactic acid bacteria, and Enterobacteriaceae. Bacterial growth kinetics order was: air > VP > MAP. *Shewanella putrefaciens* and *Pseudomonas* spp. are the specific spoilage bacteria of iced fresh fish and of refrigerated fish stored in air, but their population decreases in the absence of oxygen, *Pseudomonas* spp. being strictly aerobic bacteria. The higher growth rate of LAB under MAP conditions compared to other species likely results from their tolerance of carbon dioxide [55]. Modified atmosphere packed marine fish from temperate waters and from fresh or tropical waters have been reported to be spoiled by CO_2_-resistant Gram-negative *Photobacterium phosphoreum* and by Gram-positive bacteria, respectively [53]. Interestingly, differences regarding the nature of off-flavors and off-odors also depend on the origin of the fish: when stored in air, spoilage of fishes from temperate waters results in offensive fishy, rotten, H_2_S off-odors, while fruity and sulfhydryl off-odors are detected following the spoilage of some tropical or freshwater fishes. Broekaert et al. [54] reported a correlation between the formation of volatile compounds such as trimethylamine, ammonium, and H_2_S and specific spoilage organisms such as *Shewanella* sp., *Pseudomonas* sp., and *Photobacterium phosphoreum*. Trimethylamine results from the decomposition of trimethylamine oxide (TMAO), which is present in all marine fishes. TMAO is part of non-protein nitrogen (NPN), which is present in large amounts in fishes and promotes the growth of microorganisms requiring NPN for their growth, such as lactic acid bacteria. In vacuum-packed or modified atmosphere without oxygen packed fishes, trimethylamine oxide can be used as the terminal electron acceptor in an anaerobic respiration by specific spoilage organisms such as *Shewanella putrefuciens*, *Photobacterium phosphoreum*, and *Vibrionaceae* and result in trimethylamine accumulation. Since *Photobacterium phosphoreum* requires sodium, it does not play a significant role in the spoilage of vacuum-packed freshwater fishes. Unlike for meats, unfortunately, packing fishes in a CO_2_-enriched atmosphere results in a limited extension of shelf life compared to aerobic or vacuum-packed storage, because of TMA accumulation, which is limited during aerobic storage and which is only delayed a few days compared to VP.

Hydrogen sulfide and methyl mercaptan formation has been reported to result from the decomposition of the sulfur-containing amino acids cysteine and methionine, respectively. These volatile compounds are responsible for off-odors apparition. Among sulfide-producing bacteria, *Shewanella putrefaciens* is mainly responsible for H_2_S off-odors formation in aerobically and cold stored fish from arctic and temperate waters [56].

Interestingly, Lopez-Caballero et al. [57] investigated the effect of O_2_ and CO_2_ concentrations in atmosphere on the growth and the metabolic activity of a *Shewanella putrefaciens* strain isolated from spoiled hake (*Merluccius merluccius* L.) for 3 weeks at 1 °C following its inoculation (10^4^ cfu/mL) in sterile fish juice. Air storage resulted both in the highest *Shewanella putrefaciens* population (>10^9^ cfu/mL) and TMA concentration (45 mg/100 mL) after 3 weeks and in strong putrid off-odors after 2 weeks. Contrarily, all modified atmospheres reduced bacterial growth, TMA, off-odor and biogenic amines formation, and a 40% CO_2_ and 60% O_2_ mixture had the highest inhibitory effect on bacterial growth.

Dalgaard [58] reported that while sulfidy off-odors were detected in refrigerated whole cod stored aerobically, which was not the case for VP and MAP chilled spoiled cod fillets. Off-odors of VP and MAP cod fillets were due to TMA and ammonia-like off-odors. They compared TMA-production of *Shewanella putrefaciens* and *Photobacterium phosphoreum* cells in fish juice: *P. phosphoreum* cells produced 30 times more TMA than *S. putrefaciens* cells. Since TMA concentration in spoiled packed cod fillets (30 mg/100 g) is consistent with *P. phosphoreum* population enumerated (10^7^ cfu/g) and since only low levels of *S. putrefaciens* were found, they concluded that *P. phosphoreum* was responsible for TMA formation. Moreover, the high CO_2_-tolerance of *P. phosphoreum* might promote their viability in packed fishes.

Several authors recently inoculated sterile fish fillets with different strains of specific spoilage bacteria and monitored the degradation pathways of fish flesh constituents (e.g., glucose, total sugars (including glycogen), proteins, non-protein nitrogen fraction) during refrigerated storage (e.g., [59,60]). For instance, Yi and Xie [59] inoculated bigeye tuna fish blocks with two *S. putrefaciens* strains before 10 days storage at 4 °C. While the growth patterns of the two strains, from which population reached ~10^9^ cfu/g after 10 days, were similar, TVB-N and TMA increased far more rapidly following inoculation with one of the two strains. Interestingly, a comparative proteomic analysis of intracellular and extracellular proteins between both strains and biochemical analyses allowed to propose that the strain inducing a more rapid spoilage of tuna fish flesh also had a higher extracellular protease activity and that its upregulated proteins were mainly involved in amino acid, sulfur, and carbohydrate metabolisms. Differences in deamination and decarboxylation activities between the two strains were also pointed out by the same authors in a more recent study [59]: they suggested that it also contributed to the higher ammonia and cadaverine or putrescine production, respectively. Nevertheless, while this kind of study provides relevant information for a better understanding of fish spoilage mechanisms by different bacterial strains, the effect of packaging atmosphere, which affects both microbial growth and metabolism, has rarely been considered to date. Zhuang et al. [60] inoculated grass carp flesh with *Pseudomonas putida*, *Aeromonas rivipollensis*, or *Shewanella putrefaciens* strains. While the growth pattern over 15 days storage under aerobic conditions of the three strains was similar, *S. putrefaciens* enhanced putrescine and cadaverine production unlike *A. rivipollensis*. As expected, *P. putida* had the highest proteolytic activity but also the highest deamination activity, which would explain the highest ammonia production following its inoculation.

For more details regarding correlations between storage conditions (including temperature and packaging conditions), specific spoilage microorganisms, and volatile organic compounds, the interested reader can refer to Odeyemi et al.’s [61] review.

## 4. Volatilome Analysis

### 4.1. TVB-N and TMA

Two conventional analysis methods related to volatile compounds: TMA (AOAC. (1990). Official methods of analysis of the AOAC, 15th ed. Methods 932.06, 925.09, 985.29, 923.03. Association of Official Analytical Chemists. Arlington, VA, USA.) and TVB-N [62] (a global indicator) are used as a common index of spoilage [48,63,64]. A recent review [18] focused on analysis methods used to determine fish freshness. However, considering TMA and TVB-N as the major volatile compounds formed in stored fish is very reductive.

Various limits of acceptability of TVB-N to estimate fish freshness can be found, levels above 35 mg N/100 g always signifying that the fish is spoiled. Examples of levels fixed by the European Commission are stated in Table 2.

Taliadourou et al. (2003) proposed 26.88 mg TVB-N per 100 g flesh as limit of acceptability for filleted fish, 26.77 mg per 100 g for whole ungutted sea bass [63], and Ozogul et al. [65] proposed 10 mg per 100 g for European eel (*Anguilla anguilla*). Thus, it appears that defining a limit for TVB-N value is not so simple. Furthermore, as reported in Section 3, TVB-N level cannot be used as an early stage of fish degradation indicator since values are not relevant before at least 10 days of storage. It has also been shown that environmental conditions are important for TVB-N values evolution during long storage periods. Castro et al. [66] monitored the changes of TVB-N level along fish storage and indicated that there were no significant changes during the edible storage life before 21 days and increasing changes after 21 days of storage. Sivertsvik et al. [35] reported that TVB-N was decreasing when fishes were stored under CO_2_ atmosphere, even in the case of bad sensory evaluation. This suggests thus that TVB-N assay is not always relevant to qualify spoilage of modified atmosphere packaged fish.

There is no regulation concerning the TMA level but beyond 12 mg-N TMA per 100 g, the product quality is generally considered damaged [12]. Nevertheless, while TMA is an interesting indicator under aerobic conditions, TMA is a reaction intermediate under anaerobic conditions, which prevents it from being a marker in these conditions.

These methods are informative, but do not precisely characterize freshness and are not effective for all species. There is thus a necessity to find more specific methods for fish freshness/spoilage monitoring. Volatile compounds (contained in packaging headspace or extracted by techniques such as SPME) are good indicators of fish freshness. More precisely, while some of them (TMA, acetic acid, etc.) are increasing during storage and can be connected to food decomposition and/or microbial development others, such as carbon disulfide, which can be associated to specific aroma of fresh fishes, are decreasing [39]. Thus, a global analysis of volatilome will be more informative than a simple TMA analysis by conventional techniques. 

### 4.2. Volatilome Analysis and Identification of Specific Markers

The non-targeted identification of VOCs along packed fish refrigerated storage can result in the identification of new specific markers of its quality. However, specific markers of fish quality such as compounds responsible for off-odors may result from compounds at trace levels with a low odor threshold necessitating highly sensitive methods for their detection [14].

Since VOCs are diluted in the headspace atmosphere of packed fish, direct analysis of headspace atmosphere (static headspace) is often not sensitive enough and a VOCs preconcentration step is often needed prior to their analysis. Several methods using solvents, such as solvent-assisted flavor evaporation (SAFE) [67] or liquid–liquid extraction (LLE) [68], are used to recover VOCs. However, these are not commonly used methods. Generally, the VOCs are adsorbed on a solid phase like for solid phase micro-extraction (SPME), which is a static method with limited adsorption capacity of the adsorbing fiber due to the thickness of the coating materials and the short length (~1 cm) and diameter (~100 μm) of the fibers used [69]. That is why other methods with a higher adsorption capacity, such as monolithic material sorptive extraction (MMSE) [69], stir bar sorptive extraction (SBSE), or SPME arrow, were developed. The advantage of SPME arrow is that the analyses can be automated, while MMSE and SBSE are manual.

Another strategy to improve the sensitivity is to entrain the volatile molecules of interest using a flow, it is the dynamic headspace (DHS) sometimes called “purge and trap”. The temperature of extraction using SPME or DHS is often relatively high, 40 °C [48,70,71], 50 °C [47,49,51,72], 60 °C, or more [73,74] and the time of extraction ranges from 15 min to 2 h. To our knowledge, only Olafsdottir et al. (2005) and El Barbi (2007) [75,76] did the extraction at 23 °C or room temperature. To reduce the temperature and time of extraction some authors suggest using vacuum [77,78,79]. It has not been tested on fish, as far as we know. Zhang et al. (2022) [71] proposed a comparison of extraction methods, SPME, SAFE, DHS, SBSE, and LLE. The OPLS-DA (Orthogonal Projections to Latent Structures Discriminant Analysis) analysis showed that the number of odors extracted was higher following SPME. A higher level of 1-octen-ol was detected, which is why the final choice was SPME [68]. It is possible to inject directly the headspace into a gas chromatograph, in this case there is no concentration of molecules and the risk of losing information, notably for molecules in small concentration.

As recently reviewed by Epping and Koch [80], VOCs identification and quantification are also challenging namely due to their chemical diversity in addition to their low concentration. The most common measuring method for VOCs is gas chromatographic (GC) separation followed by different kinds/types of detection, mass spectrometry being the gold standard. Other methods used to separate the VOCs are: Selected Ion Flow Tube (SIFT) and Secondary ElectroSpray Ionization (SESI) (more complex but with a higher sensitivity compared to SIFT [81]). Several isomeric compounds may have the same SIFT-MS analyte ions, (e.g., 3-ethyl-1-butanol/pentanol, 3-methyl butanoic acid/pentanoic acid, and 3-methylbutanal/pentanal) [81], which constitutes a bias. Moreover, after separation, Olfactometric (O) analyses can also be performed [68]. There is only one publication with quantification spiked with analytical standards solutions [72], others compared peak area or area ratio with internal standards. This illustrates the difficulty of quantifying volatile molecules with repeatability issues, saturation of the trap, etc. Authors such as Olafsdottir et al. (2005) reported that the variation following GC analysis of duplicate samples was high especially for the very volatile compounds, such as acetaldehyde and ethanol [75]. 

Major VOCs identified during fish refrigerated storage in the literature following suitable extraction and identification methods are listed in Table 3. Some VOCs have been identified as a key in sensory rejection by GC-Olfactometry. Especially ketones, mainly 3-hydroxy-2-butanone, TMA, and some alcohols [67,75]. Acetoin and 2- and 3-methyl-1- butanol were proposed as potential fish quality markers: l [82]. Indeed, levels of acetoin increased earlier than TMA; therefore, it is more relevant to monitor the loss of freshness as an early indicator of spoilage [75] by monitoring such VOCs. Some authors suggest that species-specific markers of spoilage will need to be defined [49], or different markers depending on the storage conditions [32,83].

To be an adequate indicator of spoilage, the molecule must be produced early and in sufficient quantity to be assayed. This is why some authors ruled out some molecules: in the condition studied by Emborg et al. (2002) [84], methyl mercaptan and dimethyl trisulfide aldehydes accumulated when the products were already spoiled [75], while 3-methyl-1-butanol, 3- and 2-methylbutanal, and ethyl esters of short chain fatty acids (C4-–C10) were produced (if any) at very low levels during storage.

Tanimoto et al. (2020) developed a method for the rapid screening of the effect of plant extracts on the quality deterioration of dark muscle fish flesh after 3 days storage at 4 °C. Their method was based on the monitoring of color change (browning) and of four volatile compounds (propanal, 2,3-pentanedione, hexanal, and 1-penten-3-ol). A total of 11 out of 24 plant extracts significantly reduced color browning as well as the production of these four compounds. Interestingly, the five most efficient plant extracts regarding inhibition of the four volatile compounds production were confirmed by sensory evaluation to effectively reduce dark muscle fish flesh deterioration [85].

Other molecules, such as 1-hexanol-2-ethyl [49,71], piperidine [47,75], heptanone [49,74] heptenal [48,74], and 2-methyl 1-propanol [49,75] have also been studied but less often cited than those listed in Table 3.

**Table 3 foods-12-02657-t003:** Major VOCs in packed refrigerated fish identified in the scientific literature.

Class	VOC	Extraction and Analytical Method	References
Acids	Acetic acid	SPME/GC/MS	[13,48,51,70,73]
		Dynamic headspace/GC/MS	[47,75]
		Static headspace/SIFT/MS	[81,86]
Alcohols	Ethanol	SPME/GC/MS	[46,48,49,70]
		Dynamic headspace/GC/MS	[75]
		Static headspace/SIFT/MS	[32,81,86]
	3-methyl-1-butanol	SPME/GC/MS	[49,51,72,76]
		Dynamic headspace/GC/MS	[47,75]
		Static headspace/SIFT/MS	[32]
	2,3-butanediol	SPME/GC/MS	[49,70]
		Dynamic headspace/GC/MS	[75]
		Static headspace/SIFT/MS	[68,86]
	1-penten-3-ol	SPME/GC/MS	[48]
		Dynamic headspace/GC/MS	[47,74,75]
Aldehydes	2 or 3 methyl-butanal	SPME/GC/MS	[48,49,51]
		Dynamic headspace/GC/MS	[75]
		Static headspace/SIFT/MS	[32]
Amines	TMA	SPME/GC/MS	[49,72,76]
		Dynamic headspace/GC/MS	[47,75]
Esters	Ethyl acetate	SPME/GC/MS	[49,71]
		Dynamic headspace/GC/MS	[75]
		Static headspace/SIFT/MS	[32,86]
Ketones	3-hydroxy-2-butanone (=acetoin)	SPME/GC/MS	[49,51,70,72,73]
		Dynamic headspace/GC/MS	[82]
		Static headspace/SIFT/MS	[32,86]
Sulfur compounds	Hydrogen sulfide, carbon disulfide and dimethyl disulfide, methanethiol (methyl mercaptan)	SPME/GC/MS	[76]
		Dynamic headspace/GC/MS	[47,75]
		Static headspace/SIFT/MS	[32,81,86]

### 4.3. Electronic Nose

The electronic nose (EN) technique has also been used to monitor the spoilage of fish, to detect changes in signal patterns during storage [87,88,89]. Hindle et al. used THz waves to monitor the production of H_2_S during refrigerated storage of Atlantic salmon under 100% N_2_ [90]. Olafsdottir et al. (2005) suggested including selective sensors in the electronic nose for the detection of ketones and acids to monitor the spoilage of cod fillets in addition to a more sensitive sensor for the detection of TMA [75]. Natale (2001) combined 2 EN to get good classification performances regarding fish freshness: only 4% of 72 fish samples stored at 10 °C for up to 17 days were misclassified, but the errors are not negligible because days 7–9 were classified as day 1 [91]. Some authors built EN using metal oxide gas sensors [76,92,93]. This technique provides interesting results with a 94% success rate of correct fish freshness evaluation, EN, as the NeOse Pro sensor (Aryballe technologies, Grenoble, France) [75,87,88,91,94], may provide a viable approach to determine fish freshness, which could be used for quality control and inspection purposes. EN must be simple instruments for real applications in the food sector [88].

## 5. Effect of Different Factors Affecting Fish Quality on Volatilome Composition

### 5.1. Effect of Temperature

In their study, Du et al. [87] showed that salmon fillet spoilage by bacteria (estimated to correspond to a total aerobic viable count (TVC) exceeding 7 log CFU/g) was reached in only 4 days at 10 °C compared to 12 days at 4 °C (initial TVC was around 2 log CFU/g). Moreover, after 7 days of storage, histamine was detected in 25% and 50% of samples placed at 4 °C and 10 °C, respectively. After 10 days, histamine was found in all samples stored at 10 °C. These samples were also submitted to a sensorial panel, which qualified salmons as unacceptable after 3 days and 14 days for samples stored at 10 °C and 4 °C, respectively. For fish samples stored in air, sensory analysis was thus a better criterion of fish spoilage than histamine determination.

Miks-Krajnik et al. [72] also investigated raw salmon fillet deterioration under aerobic conditions for up to 14 days at 4, 10 °C and up to 3 days at 21 °C. Maximum TVC was around 9.2–9.4 log CFU/g at 4 and 10 °C and 8.8 log CFU/g at 21 °C. The lower TVC at 21 °C was assigned to a lower growth of psychotropic bacteria. Predicted shelf life, defined by a TVC upper limit of 7 log CFU/g, was achieved in 3.02, 1.82, and 0.6 days at 4, 10, or 21 °C, respectively. Differences compared to Du et al.’s [87] experiment can be ascribed to a higher initial TVC (5.1 log CFU/g). Therefore, even if storage temperature is an important factor for fish spoilage prevention, initial microbial load is a predominant parameter. Miks-Krajnik et al. [72] also observed that specific spoilage bacteria enumeration, such as Pseudomonas or H_2_S-producing bacteria, did not result in a better prediction of shelf life than the values derived from TVC. In addition to microbial counts, these authors also measured the sensory acceptance of samples stored at 4 °C by untrained panelists considering color, odor, and texture. Consistently with Du et al. [87], they also observed that sensory rejection can be correlated with microbial growth. To complete these data, they measured the volatile content of salmon samples headspace: they observed that the main compounds which may be used as spoilage indicators are acetoin, ethanol, acid acetic, TMA, and 2,3 butanediol and that the indicators profiles are temperature-dependent.

Alfaro et al. [40] studied the quality changes of MAP (48% CO_2_, 50% N_2_, 2% O_2_) horse mackerel fillets stored at temperatures from 2 to 10 °C. The higher the temperature, the more rapidly the sensory indicators (odor, color, muscle firmness) were affected (from within 7 days at 2 °C to within 3 days at 10 °C). Sensory analysis was correlated with TVC and psychrotrophic bacteria count (bacterial load correlated with sensory alteration was 10^6^ CFU/g). Specific volatile compounds (acetaldehyde and butyraldehyde) were detected in the packaging atmosphere only at 10 °C. No lipid oxidation was observed. Moreover, TVB-N values indicated food spoilage except at 2 °C. Concerning TMA, no production was observed at low temperatures (i.e., 2 and 4 °C) even after 11 days of storage. In addition, these authors observed a significant increase of three aldehydes (acetaldehyde, 2-butanone, and butyraldehyde) during fish storage at 10 °C. It is noteworthy that butanone and butyraldehyde accumulations were correlated with advanced spoilage state.

These three studies pointed out that fish storage at low temperatures leads to an extended shelf life, but that it is difficult to determine a simple biochemical indicator for fish deterioration. Due to differences in fish species, packaging systems (aerobic, VP, MAP), and storage conditions (mainly temperature and duration), it is not easy to determine the main tendencies of fish quality evolution during MAP storage [7]. As expected, the rate of deterioration of fish is highly temperature-dependent and can be inhibited using low storage temperatures [35].

### 5.2. Effect of Atmosphere Composition

VP or MAP result in a limited shelf life extension of seafood products compared to meat, due to CO_2_-resistant bacteria (namely *P. phophoreum* as previously stated) growth and metabolism [84,95]. As a result, studies focused on the identification of chemical indicators to characterize fish spoilage [82], when packed under anaerobic and/or modified atmosphere with CO_2_ conditions. As already described in Figure 1, the initial headspace composition of packaged (AP, VP, and MAP) fish evolves by oxygen depletion and CO_2_ production. For example, Narasimha Rao et al. [96] considered VP as a type of MAP since the consumption of the residual oxygen of the packs by microorganisms results in the production of carbon dioxide within the package.

For fish matrices, MAP composition evolves over time [26,97] and to our knowledge, no Equilibrium Modified Atmosphere Packaging (EMAP) was reported to date, unlike for respiring products such as fresh vegetables and fruits. Usual modified atmospheres (or MAP) are called CO_2_-MAP for MAP enriched in CO_2_ (N_2_ + CO_2_ blends) or CO_2_-O_2_ MAP for ternary blends (CO_2_ + O_2_ + N_2_). Nitrogen is a filling gas, used to modulate the proportions of the other gases present in the packaging. Its low solubility in water and fat prevents package collapse and deformation caused by the comparatively far higher CO_2_ dissolving into the food tissues [98,99].

By replacing O_2_, CO_2_-MAP plays an important role in delaying oxidative rancidity apparition and inhibits the growth of strictly aerobic pathogenic and spoilage microorganisms. Many articles reported the extension of the shelf life of seafood products resulting from CO_2_-MAP [23,100]; Garcia-Gonzalez et al. (2007) and Yu and Chen (2019) [101,102] reviewed the mechanisms of carbon dioxide bactericidal action, introducing the use of carbon dioxide as an alternative cold pasteurization technique for foods, reported to inactivate both microorganisms and enzymes [103]. In contrast, the organoleptic properties of fish can be altered: lower firmness and changes in color were observed when filleted Atlantic salmon was packed using a CO_2_ emitter, or traditional MAP [34,35]. Indeed, the CO_2_ dissolution in the aqueous phase of fish flesh tends to decrease the pH up to a 0.2–0.3 lower value depending on the CO_2_ concentration in the surrounding atmosphere. For example, Emborg et al. [84] considered that the slightly longer shelf life of salmon could be partly explained by the lower pH (~6.3) in salmon compared to white-flesh fish (~6.6). Differences in spoilage microflora and pH are thus primarily responsible for the shelf life of packaged foods and therefore volatile chemical indicators. 

It is difficult to identify degradation indicators induced by anaerobic conditions (VP) and atmospheres enriched in CO_2_ or O_2_ and CO_2_ (MAP). For CO_2_ for example, the concentration in the package is not always controlled because of the dissolution of CO_2_ in the fish flesh and concomitant temporary CO_2_ depletion [32] before its accumulation resulting from microbial respiration: the outflow of CO_2_ from packaging is then variable, depending on the instantaneous pressure gradient, established between the inside and the outside of the package. Comparatively to CO_2_, monitoring oxygen concentration evolution is easier: oxygen depletion results from microbial respiration (and from oxidation to a lesser extent). The oxygen flux entering in the packaging depends on its permeability. These permeability data are, however, sometimes missing or partially described (as well as the packaging geometry), the packages being sometimes described only by their commercial names.

Nevertheless, Kuuliala et al. [32] monitored the evolution for 14 days of the headspace gas concentrations (CO_2_/O_2_) throughout storage time for different initial atmosphere conditions, in order to try to evaluate the spoilage of raw Atlantic salmon based on the VOC profile. This allowed them to identify the main spoilage indicators. Under anaerobic conditions, ethanol, dimethyl sulfide, and H_2_S were found to be characteristic irrespective of the applied CO_2_ levels, varying from 0 to 50% (*v*/*v*). Under air, most of the identified VOCs were alcohols and ketones. Fewer compounds were identified under high-CO_2_ MAP (50%) when compared to low-CO_2_ conditions (10–20%) and 3-methylbutanal was the only identified VOC under aerobic MAP (40–50% O_2_). In conclusion, ethanol, dimethyl sulfide, H_2_S, methyl mercaptan, and acetoin were found abundant under several conditions and often identified as main spoilage indicators, suggesting that monitoring these VOCs in the package headspace could lead to significant benefits in the seafood industry.

In another study, Oluwole [104] identified 16 volatile compounds in the headspace of Cape hake fillets when packed in a 40% CO_2_, 30% O_2_, 30% N_2_ atmosphere, despite a systematic higher drip loss: the authors concluded the necessity of using an absorbent pad in addition to MAP refrigerated storage to maintain the firmness of fillets. The VOCs associated with spoilage include trimethylamine (TMA), esters, and sulfur groups. MAP had a significant influence on volatile composition and concentration. For example, TMA concentration was only 0.85% on day 12 compared to 7.22% on day 6 at 0 °C under air (the authors named this condition a passive MAP). However, other authors considered TMA an irrelevant indicator for salmonids because of the low amount of endogenous OTMA in these fishes [84,105]. 

Nevertheless, Gokoglu [100], in his review dedicated to seafood packaging, reported the upcoming use of ternary MAP (i.e., CO_2_-O_2_ MAP with low O_2_ levels) to prevent both the reduction of TMAO to TMA and anaerobic bacteria growth. Erickson et al. [24] also discussed the use of O_2_ semi-permeable packaging to ensure a progressive O_2_ ingress in the headspace of packaging. If packed fish is exposed to temperature abuse, O_2_ presence could prevent the growth and toxin production by *C. botulinum*, monitored before spoilage and possibly detected by the analysis of headspace volatile compounds (increase in a peak on GC-chromatogram coincided with the formation of toxin).

Acetoin and 2- and 3-methyl-1-butanol have often been reported as relevant and reliable fish spoilage markers [13,49,73,106], Erickson et al. [82] underlined that ideal spoilage indicators should demonstrate clearly increasing or decreasing concentrations with storage time. However, for some volatile microbial metabolites, concentrations may increase temporarily, reach a plateau, and then decrease as the dominant species of spoilage bacteria changes over time. Several authors found these tendencies for most of volatile compound peaks detected during the storage of salmon with different packaging films and atmospheres at different temperatures. This phenomenon is due to (i) the solubilities of CO_2_ and volatile compounds in food matrices, which are temperature dependent as well as microbial and enzymatic activities [107], and (ii) the instantaneous flux of O_2_ and CO_2_ (and volatile compounds) across the film that is a function of the permeabilities of the film to these gaseous molecules and their concentration gradients, which also depend on their interaction with foods through a reaction diffusion mechanism difficult to predict in such dynamic systems [30].

Taken together, the results of the different studies demonstrated that changes in volatile compounds were significantly influenced by storage duration, temperature, and MAP, but also by the initial microbial ecosystem and microflora loads present onto the food and materials packaging which are not always precisely described.

### 5.3. Hurdle Technology

Leistner (1985) [108] proposed that each factor/technology used to control microorganisms to extend shelf life and/or improve safety of foods can be seen as a “hurdle” to be overcome by unwanted microorganisms. Hurdle technology derives from the understanding of this “hurdle effect” by considering that when different food preservation technologies are combined, the level of preservation activity provided by one technology can be increased additively or synergistically. As reviewed by Leistner (2000) [109], progress in understanding of the mechanisms by which each technology acts on unwanted microorganisms offers the possibility to apply an intelligent mix of hurdles acting simultaneously on different targets within the microbial cells, thereby favoring a synergistic preservation effect. This multi-target preservation strategy is particularly promising for the preservation of fresh or minimally processed fish and seafood, which are prone to rapid deterioration of their quality. Indeed, combining modified atmosphere packaging and refrigeration already results from hurdle technology principles application, since for instance low temperature and reduced oxygen content, resulting also in a modified redox potential (Eh), are factors that can have different effects on microorganisms. Nevertheless, in the context of this review, where seafood packaging technologies effect on the volatilome has already been examined, this part of the review is focused on the effect on volatilome of the combination of packaging technologies, including MAP and VP, with technologies such as (i) non-thermal treatments, (ii) the application of antimicrobial and antioxidant compounds, or (iii) the use of bioprotective microorganisms. While there is an ever-increasing number of studies regarding the combination of these technologies with refrigeration and MAP, studies considering the effect of these combinations on volatilome composition and evolution are scarce and less specific indicators of fish quality such as TVB-N or trimethylamine assay were monitored in a limited number of studies listed in Table 4. In addition, the application of non-thermal treatments and of alternative chemical compounds including essential oils and bacteriocins to the preservation of fish and meat has been recently reviewed by Rosario et al. (2021) [110].

#### 5.3.1. Modified Atmosphere or Vacuum Packaging in Combination with Non-Thermal Treatments (High Hydrostatic Pressure, Irradiation, UV-C Treatment, Treatment with Ozonated Water, etc.)

The application of non-thermal treatments to seafood preservation, such as HHP, ionizing radiation, cold plasma, ultraviolet light, and pulsed electric fields, have been reviewed by Olatunde and Benjakul (2018) [111]. Before combining non-thermal treatments with modified atmosphere or vacuum packaging, the effect of various non-thermal treatments applied alone on the evolution of fish quality during post-treatment refrigerated storage was monitored in several studies [38].

HHP-treated (200–600 MPa) vacuum-packed fish fillets with an extended shelf life have already been launched by several companies in the world including Delpierre (Nantes, France). Günlü et al. (2014) [112] also reported an extended shelf life of vacuum-packed rainbow trout fillets treated for 5 min at 220 MPa at 15 °C. However, HHP treatment had less of an effect on both shelf-life extension and TVB-N increase reduction than coating of the same fillets in chitosan-based films. This might be related to the fact that a 220 MPa HHP is not very high: indeed, the pressures required to achieve microbial inactivation are usually in the 300–700 MPa range.

Reale et al. (2008) [113] monitored the quality of seabasses packed in polyethylene film bags directly in a normal atmosphere (control) or following a 3 kGy γ-irradiation treatment to reduce or eliminate unwanted microorganisms, packed in a 40:40:20 CO_2_/N_2_/O_2_ gas mixture, or packed in a 60:35:5 CO_2_/N_2_/O_2_ gas mixture. Interestingly, the monitoring of fish quality evolution during refrigerated storage at 2 °C for up to 10 days comprised a sensory evaluation including evaluation of gills and internal odors of seabasses. While fillets stored in a MAP still had seaweed and shellfish odors after 4 days, this odor was less sharp for control fillets and only just detectable for ionized fillets. Estimation of freshness considering only odor evolution thus leads to the following ranking: MAP > ionization treatment > control. It would have been interesting to examine the combination of an ionization treatment followed by MAP.

Lazaro et al. (2020) [114] compared the effects of MAP in a 50% CO_2_, 50% N_2_ atmosphere preceded or not by UV-C treatment of Tilapia fillets with air-packed or VP fillets on pathogens reduction and on their shelf life. Interestingly, ammonia concentration in fillets after 10 days storage at 4 °C was lower following UV-C treatment and MAP alone or combined than when fillets were air- or vacuum-packed. 

Non-thermal plasma treatment results in the formation of reactive species including reactive oxygen radicals, positive and chemical ions, excited molecules, and electrons. Unfortunately, these reactive species not only result in microbial inactivation, but also promote lipid oxidation in fish. Therefore, Tagrida et al. (2021) [115] combined non-thermal plasma treatment of Nile tilapia fillets with betel leaf extract addition. Antioxidant compounds of betel leaf extract thus limited lipid oxidation.

#### 5.3.2. Modified Atmosphere or Vacuum Packaging in Combination with Antimicrobial and/or Antioxidant Compounds

Fresh seafood being prone to microbial spoilage and to oxidation of its lipids because of its high content of highly polyunsaturated fatty acids, addition of antimicrobial, and/or antioxidant compounds to extend its shelf life has been investigated by many authors. Antioxidants such as ascorbic acid and its salts or preservatives such as potassium sorbate, which are food additives, can be used. Nevertheless, concerns regarding the innocuity of some synthetic food additives has stimulated the search for alternative natural sources of antimicrobial and/or antioxidant molecules, which could be used for seafood preservation [116]. In this context, some edible plants extracts containing volatile (e.g., essential oils) and/or non-volatile antioxidant and/or antimicrobial molecules (e.g., phenolics, alkaloids) are promising. When volatile antimicrobial molecules are used, they can be added in the headspace of packaging system. Their respective concentrations in the headspace of food packaging and in the food matrix will namely depend on their vapor phase-food matrix partition coefficient. Solubility of volatile molecules in food matrices will namely depend on their composition (e.g., fat content). Moreover, the permeability to each active volatile molecule of packaging system will have to be determined. The necessity to consider these parameters for the design of effective packaging systems containing antimicrobial volatile molecules has been recently illustrated by Bahmid et al. [117], who determined the effective mustard seeds quantity to release a sufficient allyl isothiocyanate quantity to effectively inhibit *Pseudomonas fragi* growth in a liquid medium placed in a closed packaging system. The capacity of plant extracts to inhibit bacterial growth and biogenic amines formation during refrigerated storage of seafood has been reviewed by Houicher et al. [118]. As recently reviewed by Rathod et al. [119], microbial metabolites such as bacteriocins, reuterin, or organic acids can also exert an interesting preservative action on seafood.

The possibility of using animal origin antimicrobial molecules such as proteins (e.g., hen egg white lysozyme, [120]), peptide fragments resulting from their hydrolysis [121] or chitosan [122] to extend shelf life of seafood has also been investigated by several authors. Interestingly, chitosan is produced commercially by alkaline deacetylation of chitin from crustaceans, such as crabs and shrimps. Chitosan which possesses interesting film-forming properties and antimicrobial activity is thus a byproduct of some seafood.

Solutions/suspensions of antimicrobial and/or antioxidant molecules can be sprayed on the surface or seafood can be immersed in such solutions/suspensions before vacuum or modified atmosphere packaging. Similarly, treatment of seafood with film-forming solutions or suspensions can result in their coating. Such coatings can favor a controlled release of antimicrobial or antioxidant molecules to the surface of seafood. Interestingly, Zhang et al. [68] recently reported that grass carp fillets packed in PLA films grafted with lysozyme or chitosan had a 1 day and 3 days longer shelf life, respectively, than fillets packed in control PLA films. Such non-migrating active antibacterial packaging prepared by covalently immobilizing lysozyme or chitosan on the surface of plasma treated PLA films could provide long term antimicrobial activity. Application of such films to carp fillets packaging was also shown to delay accumulation of TVB-N, which is a common indicator of fish freshness.

#### 5.3.3. Modified Atmosphere or Vacuum Packaging in Combination with Bioprotective Lactic Acid Bacteria

As already stated by Calo-Mata et al. [123] in their review, the use of protective cultures of lactic acid bacteria to extend the shelf life and/or enhance the microbial safety of aquatic food products is promising. Indeed, lactic acid bacteria inhibit the growth of unwanted microorganisms by different mechanisms including competition for nutrients, lowering of pH by producing lactic acid, and production of other antimicrobial metabolites, such as bacteriocins. Due to their presence in many fermented foods, many lactic acid bacteria are Generally Recognized As Safe (GRAS) by the US Food and Drug Administration (FDA) and are in the list of microorganisms with a Qualified Presumption of Safety (QPS) in the European Union. However, unlike in fermented foods, bioprotective lactic acid bacteria should not alter the sensory properties of foods. Interestingly, some lactic acid bacteria can grow at low temperatures and they tolerate different atmosphere compositions. Nevertheless, as can be seen in Table 3, despite the promises of this approach, the number of investigations regarding the combination of bioprotective lactic acid bacteria with modified atmosphere or vacuum packaging for the preservation of fresh fish or seafood is still limited. Some authors also reported that combining bioprotective lactic acid bacteria with modified atmosphere packaging prolonged the refrigerated shelf life of cooked shrimps [124,125] or tuna burgers [126].

Hurdle technology principles application by combining refrigeration and MAP or VP of fresh fish or seafood with non-thermal processes inactivating microorganisms or addition of ingredients such as antimicrobial or antioxidant molecules or even bioprotective lactic acid bacteria has been shown by many authors to significantly extend their shelf life and/or enhance their safety. The existence of an operational window for applying non-thermal treatments, such as HHP treatments not inactivating lactic acid bacteria [127,128], or for adding plant extracts at sub-inhibitory concentrations for bioprotective lactic acid bacteria [129,130] opens the perspective to identify synergistic combinations of treatments that could be combined with refrigeration and MAP to further extend the shelf life or enhance the safety of fresh fish or seafood. Despite its promises, such combinations have not been tested so far for seafood preservation to our knowledge.

**Table 4 foods-12-02657-t004:** Examples of treatments combined with VP or MAP to extend the shelf life of fish or seafood according to hurdle technology principles and listing of quality parameters related to volatile organic compounds which were monitored.

Combined Treatment	Seafood/Fish Type	Quality Parameter Related to Volatile Organic Compounds Monitored	Reference
Non-thermal treatments			
HHP treatment after VP of fish fillets or of fish fillets coated in chitosan-based edible films	Rainbow trout (*Oncorhynchus mykiss* Walbaum) fillets	-TVB-N assay every 4 days up to 44 days storage at 4 °C. -The acceptable limit value of 30 mg TVB-N/100 g has been exceeded in the VP, VP HHP-treated, coated in chitosan-based film VP, and coated in chitosan-based film VP HHP-treated fillets on the 12th, 20th, 24th, and 44th days storage at 4 °C, respectively. Coating in chitosan-based films and to a lesser extent HHP-treatment decreased TVB-N increase during storage. Combination of coating in chitosan-based film and subsequent HHP-treatment have a synergistic effect on the decrease of TVB-N increase during refrigerated storage of trout fillets.	[112]
UV-C treatment before MAP (50% CO_2_-50% N_2_)	Nile tilapia (*Oreochromis niloticus*) fillets	-Ammonia concentration in fillets after 10 days storage (AP, VP, MAP, UV-C treated and AP, UV-C treated and MAP). -Ammonia concentration after 10 days storage at 4 °C was lower in MAP and UV-C treated fillets or fillets UV-C treated before being packed in a MAP.	[114]
Ozonated water (0.3 mg.L^−1^) and MAP (50% CO_2_-50% N_2_)	Striped red mullet (*Mullus surmuletus*)	-Assay of TVB-N, TMA-N and odor of raw fish at regular intervals. -TVB-N and TMA-N limit values were reached later when raw fish samples were treated with ozonated water prior to MAP. Nevertheless, the limit of acceptability based on sensory analysis was reached after 10 days storage at 4 °C on both cases (vs. after 7 days for control samples stored in air).	[131]
Treatment by cold plasma generated using the mixed gases (oxygen, carbon dioxide argon: 10: 60:30) for 5 min in combination with ethanolic coconut husk extract in either free or liposomal encapsulated form followed by MAP (60% CO_2_, 30% Ar_2_, 10% O_2_)	Asian sea bass slices	-TVB-N and TMA assays every 3 days for 21 days storage at 4 °C -The combination of cold plasma treatment with ethanolic coconut extract extended the shelf life to more than 18 days at 4 °C and reduced both TVB-N and TMA accumulation.	[132]
Encapsulated or non encapsulated betel leaf extract and/or in-bag dielectric discharge (80 kV, 300 s) (nonthermal plasma) and/or MAP	Nile tilapia fillets	-TVB-N and TMA assays every 3 days for 15 days storage at 4 °C. Untreated control fillets and nonthermal plasma-treated fillets subsequently stored under a CO_2_/Ar/O_2_ (60%/30%/10% *v*/*v*) MAP reached the maximal acceptable TVB-N limit (35 mg/100 g according to EU regulation) within 6 days and 9 days, respectively. Addition of 400 ppm unlike addition of 200 ppm of encapsulated or non-encapsulated betel leaf extract combined with non-thermal plasma treatment and MAP allowed to delay to over 12 days this duration of storage. The same trends were observed following TMA assay.	[115]
Antimicrobial and/or antioxidant compounds			
Dipping in 1% (*w*/*w*) cinnamon essential oil in water emulsion or in marinade containing 1% (*w*/*w*) cinnamon essential oil followed by either MAP (60% CO_2_-40% N_2_) or VP	Salmon (*Salmo salar*)	-Evaluation of off-odors by sensory analysis. -Cinnamon essential oil addition did not extend microbial shelf life of salmon. -No significant effects on odor relative hedonic score evolution of salmon during 14 days refrigerated storage of VP or MAP and dipping with cinnamon essential oil or not were observed.	[133]
Dipping for 30 min in 0.3% or 0.5% (*w*/*v*) *Capparis spinosa* root extract followed by MAP (5% O_2_, 20% CO_2_, 75% N_2_)	Rainbow trout (*Oncorhynchus mykiss*) fillets	-TMA assay and sensory evaluation (including odor) after cooking at 98 °C for 20 min every 7 days up to 28 days of refrigerated storage. -Dipping in *Capparis spinosa* root extract prior to MAP significantly delayed TMA accumulation and cooked fish odor alteration and increased rainbow trout fillets shelf life.	[134]
Oregano essential oil and/or nisin combined with MAP (75% CO_2_-25% N_2_)	Grass carp (*Ctenopharyngodon idellus*)	-Assay of TVB-N and monitoring of odor every 4 days during storage at 4 °C. -Shelf life of nisin-treated and oregano essential-oil treated grass carp fillets increased from 16 days to 20 days, while it increased to 28 days following a nisin and oregano essential oil treatment. -TVB-N values evolution during storage was not modified by treatments with nisin and/or oregano essential oil. Nevertheless, TVB-N values always remained below the 25 mg/100 g upper acceptable limit.	[17]
Addition of 0.1% (*w*/*v*) thyme essential oil prior to MAP (5% O_2_, 50% CO_2_, 45% N_2_)	Fresh Mediterranean swordfish fillets	-TVB-N and TMA assays and odor sensory evaluation for 16 days storage at 4 °C. -Extension of shelf life from approximately 13 days for MAP fillets to 15.5 days for fillets added with thyme essential oil before MAP.	[135]
Addition of 2% (*v*/*w*) or 4% (*v*/*w*) sage essential oil prior to VP	Fresh rainbow trout fillets	-TVB-N and odor sensory evaluation for 34 days storage at 4 °C. -Shelf life extension by 5 and 15 days of VP fillets added with 2% (*v*/*w*) and 4% (*v*/*w*) sage essential oil, respectively (compared to VP fillets without essential oil).	[136]
Grafting of in silico designed antimicrobial peptide 1018K6 on polypropylene (PP) packaging films	Salmon (*Salmo salar*) fillets	-TVB-N and TMA assays and odor sensory evaluation for 7 days storage at 4 °C. -TVB-N and TMA accumulation as well as odor alteration of salmon fillets were lower when salmon fillets were stored in films grafted with peptide 1018K6 than in control films.	[137]
Cellulose acetate films incorporated with bifidocin A	Mackerel (*Scomberomorus niphonius*) fillets	-TVB-N assay and odor sensory evaluation over 15 days storage at 4 °C. -Addition of bifidocin A in films significantly delayed TVB-N accumulation and odor alteration.	[138]
Bioprotective lactic acid bacteria			
Dipping of gutted fishes in a 10^7^ cfu/g *Latilactobacillus sakei* suspension with or without 0.1% (*w*/*v*) glucose for 10 min prior to VP	Fresh gutted sea bass (*Dicentrarchus labrax*) and sea bream (*Sparus aurata*)	-TVB-N assay and sensory evaluation after cooking over 14 days storage at 6 °C. -Dipping in *Latilactobacillus sakei* suspension with or without glucose resulted in an extension of shelf life of both gutted fishes from 12 to 14 days and fishes dipped in *Latilactobacillus sakei* suspension with glucose had a better sensory score.	[139]
Dipping of fresh plaice fillets in a 7 log cfu.mL^−1^ suspension of *Bifidobacterium bifidium* with 400 ppm thymol for 2 min prior to air, MAP (65% N_2_, 30% CO_2_, 5% O_2_) or VP and subsequent storage at 4 °C or 12 °C	Plaices (*Pleuronectes platessa*) fillets	-Sensory analysis including odor evaluation up to 17 days of storage. -Calculated sensory shelf life appeared longer than that estimated based on monitoring of total viable count of microorganisms. -*Bifidobacterium bifidium* and thymol performed an efficient synergy in controlling hygiene indicator microorganisms.	[140]

## 6. Conclusions

Except TMA assay, the analyses of known VOCs to evaluate the freshness of fish are still scarce, despite the increased accessibility to GC-MS for VOCs identification and assay. VOCs analysis can rely upon headspace analysis but more frequently relies upon a pre-concentration of VOCs by techniques such as SPME to increase analytical sensitivity. Besides GC-MS, an increasing number of electronic noses based on an array of different selective sensors have been developed. This review also pointed out that packaging conditions and corresponding atmosphere composition evolutions during packed fish storage strongly influence the type of microorganisms contaminating fish, their growth, and their metabolic activity; this ultimately also greatly affects volatilome composition. This is well illustrated by TMA level evolution, which is not the same under aerobic or anaerobic conditions prevailing during the storage of VP products as well as of most MAP products. Nevertheless, monitoring of the volatilome of packed fishes under well-defined and controlled conditions (i.e., defined packaging system for a given fish prepared under standard hygienic conditions) allowed several authors to identify some VOCs, which are good candidates to anticipate fish spoilage and/or determine practical shelf life. A difficulty lies in the fact that individual volatilome constituents or their combinations which could be promising markers depend on the type of fish, on its origin, and on its storage conditions. Therefore, future research directions for potential spoilage volatile markers identification should include the development of a large database with the volatilome evolution of each fish species under well-defined storage conditions, which can subsequently be exploited by multivariate data analysis. The use of standard analytical techniques and conditions is also necessary to allow comparison between different studies to build a common database.

Another practical difficulty is related to the necessity to open fish packaging to sample the inner atmosphere to analyze volatilome constituents. In this context, non-invasive techniques, such as high-resolution rotational Terahertz (THz) spectroscopy are promising although its access is restricted to specialized laboratories at present [90].

## Figures and Tables

**Figure 1 foods-12-02657-f001:**
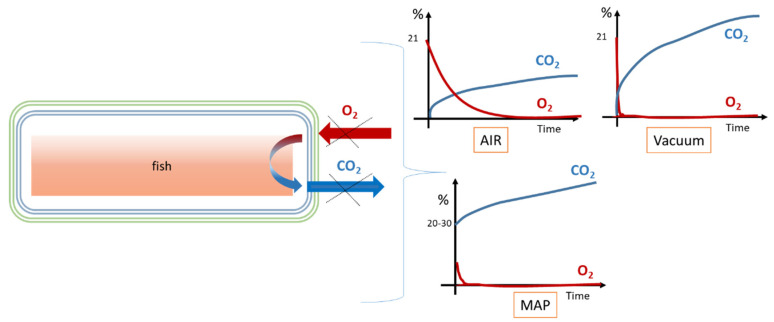
Schematic evolution over time of the atmosphere composition of the headspace for AP, VP, and MAP and for commercial gas barrier packaging (packaging with high gas barrier properties currently used for fresh fish preservation for a storage time of about 8–12 days makes O_2_ and CO_2_ exchanges negligible, especially for oxygen).

**Figure 2 foods-12-02657-f002:**
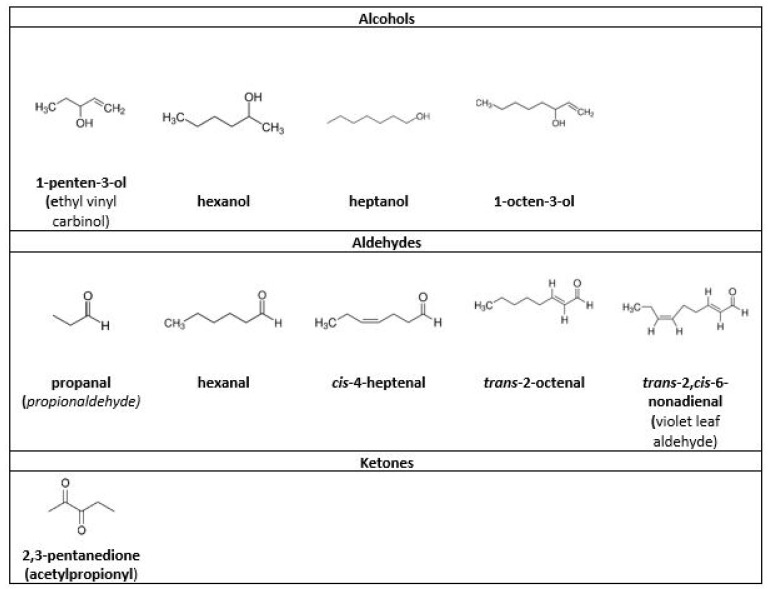
Volatilome chemicals associated with fish lipid oxidation and rancidity.

**Table 1 foods-12-02657-t001:** Examples of properties of flexible top multilayer films and of semi-rigid trays intended for the vacuum (VP) or modified atmosphere (MAP) packaging of perishable foods.

		Vacuum Packaging (VP)	Modified Atmosphere Packaging (MAP)
Flexible top film	polymer and main layer thicknesses	OPA */PE ** (15 µm/50 µm)	PE/PA ***/EVOH ****/PA/PE (global film thickness 24 µm)
	WVTR	less than 11 g/m^2^/day for 90% RH at 38 °C, ASTM E96	18 g/m^2^/day for 100% RH at 38 °C, ASTM F1249
	OTR	less than 40 cm^3^/(m^2^.day.bar) for 0% RH at 23 °C, ASTM D3985	24 cm^3^/(m^2^.day.bar) (permeance) for 0% RH at 23 °C, ASTM D3985
Semi-rigid tray	polymer and main layer thicknesses	PA/PE (80 µm/30 µm)	PET *****/PE (300 µm/30 µm)
	WVTR	less than 5 g/m^2^/day for 90% RH at 38 °C, ASTM F1249	data unavailable
	OTR	less than 50–55 cm^3^/m^2^/day for 75% RH at 23 °C, ASTM D3985	less than 5 cm^3^/tray.day (test conditions unavailable)

* OPA: Oriented PolyAmide, ** PE: PolyEthylene, *** PA: PolyAmide, **** EVOH: Ethylene Vinyl Alcohol, ***** PET: PolyEthylene Terephtalate.

**Table 2 foods-12-02657-t002:** Examples of TVB-N limits fixed by the European Commission for various fish species (adapted from European Union law 95/149/EC, 1995) [18].

Fish Species	TVB-N Limit (mg/100 g)
*Sebastes* spp. (*Helicolenus dactylopterus*, *Sebastichthys capensis*)	25
*Pleuronectidae* family (except for halibut: *Hippoglossus* spp.)	30
*Salmo salar*	35

## Data Availability

No new data were created or analyzed in this study. Data sharing is not applicable to this article.

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
