# Peer review of "Volatilome Analysis and Evolution in the Headspace of Packed Refrigerated Fish"

_foods, 2023, doi:10.3390/foods12142657_

Round 1

Reviewer 1 Report

The article itself, being a review, addresses a broader context than that described in the title in the form of a question. Perhaps a more conventional title better defines the content of the article.

The keywords should not be the same terms already described in the title, nor should they be in abbreviated form, if an acronym.

The content of Table 1 is confusing, the spacing between the columns and the presentation in the document made it difficult to understand the reported data. Maybe it was the formatting.

In the last paragraph of topic 2, line 182, it presents a conclusive content when in fact the text is still very preliminary citing few references (only 12). Maybe it would be better to rewrite it not being so emphatic in the conclusion.

Topic 4.2, line 473, presents a text with several short paragraphs, without uniformity as in the other parts of the article, generating loose and uninformative ideas.

Author Response

Dear Reviewer, dear Editor,

Please find a point-by-point response to the reviewer’s comments in the attached MSWord file or preferably see Coverletter PDF file since attached MSword file was uploaded before reviewer  2 comments.

Ø Reviewer # 1 comments and suggestions for authors

“The article itself, being a review, addresses a broader context than that described in the title in the form of a question. Perhaps a more conventional title better defines the content of the article.”

The former title (“Volatilome analysis in the headspace of packed refrigerated fish: a marker of fish degradation?”) has been replaced by “Volatilome analysis and evolution in the headspace of packed refrigerated fish.” in order to better reflect the whole content of this review, as suggested.

“The keywords should not be the same terms already described in the title, nor should they be in abbreviated form, if an acronym.”

Keywords in the revised manuscript are now: fish preservation, fish packaging, monitoring of fish quality, volatilome analysis.

“The content of Table 1 is confusing, the spacing between the columns and the presentation in the document made it difficult to understand the reported data. Maybe it was the formatting.”

Table 1 was revised in order to improve its clarity.

Title was slightly modified to state that “base semi-rigid films” correspond to “semi-rigid trays”:

Examples of properties of flexible top multilayer films and of semi-rigid trays intended for the vacuum- (VP) or -modified at-mosphere (MAP) packaging of perishable foods.

The properties of flexible top multilayer films and of semi-rigid trays are now presented in 2 different lines.

The 2 last lines of table 1 refer to the gas (water vapor and oxygen, respectively) barrier properties of top multilayer films (which have the highest permeability) and this is now stated in the table.

Revised table 1:

Please find below the revised table (without the track changes mode in order to improve readability):

Vacuum packaging (VP)

Modified atmosphere packaging (MAP)

Flexible top film

polymer and main layer thicknesses

OPA*/PE**

(15µm/50 µm)

PE/PA***/EVOH****/PA/PE

(global film thickness 24 µm)

WVTR

less than 11 g/m2/day for 90% RH at 38°C, ASTM E96

18 g/m2/day

for 100% RH at 38°C, ASTM F1249

OTR

less than 40 cm3/(m2.day.bar)

for 0% RH at 23°C, ASTM D3985

24 cm3/(m2.day.bar) (permeance)

for 0% RH at 23°C, ASTM D3985

Semi-rigid tray

polymer and main layer thicknesses

PA/PE (80 µm/30 µm)

PET*****/PE (300 µm/30 µm)

WVTR

less than 5 g/m2/day for 90% RH at 38°C, ASTM F1249

data unavailable

OTR

less than 50-55 cm3/m2/day

for 75% RH at 23°C, ASTM D3985

less than 5 cm3/tray.day

(test conditions unavailable)

*OPA: Oriented PolyAmide, **PE: PolyEthylene, ***PA: PolyAmide, ****EVOH: Ethylene Vinyl Alcohol, *****PET: Poly-Ethylene Terephtalate

“In the last paragraph of topic 2, line 182, it presents a conclusive content when in fact the text is still very preliminary citing few references (only 12). Maybe it would be better to rewrite it not being so emphatic in the conclusion.”

This paragraph has been moved to the end of text preceding Table 1 and was rewritten as follows:

As already presented in Figure 1, the headspace of packed fish evolves to a gas composition rich in CO2 and anaerobic conditions before the use-by date, since packaging with high gas barrier properties currently used for fresh fish preservation for a storage time of about 8-12 days makes O2 and CO2 exchanges negligible, especially for oxygen. »

Reference to Figure 1 better connects this conclusion to the whole content of topic 2.

“Topic 4.2, line 473, presents a text with several short paragraphs, without uniformity as in the other parts of the article, generating loose and uninformative ideas.”

This paragraph has been rewritten and 2 references ([70] and [84]) are added. We hope that it is now more informative and with a better uniformity.

Best regards

Yours sincerely

Reviewer 2 Report

The article reports interesting perspective in identifying volatile markers of packaged fresh fish with a monitor of volatile compounds' release. The article is well-written, but important concerns were found and need to be address for improvement the article. 

Figure 1: The crosses on Oxygen and Carbondioxide permeations are not clear. Instead, It is suggested to add information or words, which shows the barrier is impermeable. Impermeability means not oxygen transmission and carbondioxide throught the packaging barrier.

Information provided in Table 1 needs to improved to give clearer information for the reader. For example, OPA/PE (15/50 um) needs to be reformulated, since it is not clear what does 15/50 um mean. In the row of "flux or wvtr", the first line is "< 10-15 g/m2/day 90% RH/38 oC. It is suggested to write less than 10-15 g/m2/day for 90% RH at 38 oC. Author should reconsider "<10-15", it is better to use "<10" or "<15". In the second line, author should add unit after "<55". Please check other information in this table. Lastly, Author should add reference for each line in the table to know where the information was obtained. 

Many word extension of abbreviations are repeatable. For example TVB-N is mentioned in line 53 and 436. please check other abbreviations. 

Many paragraphs are short (<3 paragraphs), loss coherence to other paragraphs. It is suggested to combine to previous or next paragraphs. 

In the section of TVB-N and TMA (line 432), there are information related to limit of acceptability for various food products. It is recommended that the limit of accepatbility needs to overviewed in a table to know different limits for different fish products. 

In Table 2, What is "molecule" mean here? VOCs?

In Table 3, Author should shorten the information in the column of quality parameter. Take the important information

-

Author Response

Please see the attachment  or preferably see Coverletter PDF file since attached MSword file was uploaded before reviewer  2 comments.

Ø Reviewer # 2 comments and suggestions for authors

“The article reports interesting perspective in identifying volatile markers of packaged fresh fish with a monitor of volatile compounds' release. The article is well-written, but important concerns were found and need to be address for improvement the article.”

Thank you for these positive comments and your suggestions to improve the manuscript.

“Figure 1: The crosses on Oxygen and Carbondioxide permeations are not clear. Instead, It is suggested to add information or words, which shows the barrier is impermeable. Impermeability means not oxygen transmission and carbondioxide throught the packaging barrier.”

A sentence has been added at the end of Figure 1’s legend to clarify this point in the revised manuscript:

Figure 1. Schematic evolution over time of the atmosphere composition of the headspace for AP, VP and MAP and for commercial gas barrier packaging (packaging with high gas barrier properties currently used for fresh fish preservation for a storage time of about 8-12 days makes O2 and CO2 exchanges negligible., especially for oxygen).

Information provided in Table 1 needs to improved to give clearer information for the reader. For example, OPA/PE (15/50 µm) needs to be reformulated, since it is not clear what does 15/50 um mean. In the row of "flux or wvtr", the first line is "< 10-15 g/m2/day 90% RH/38 oC. It is suggested to write less than 10-15 g/m2/day for 90% RH at 38 oC. Author should reconsider "<10-15", it is better to use "<10" or "<15". In the second line, author should add unit after "<55". Please check other information in this table. Lastly, Author should add reference for each line in the table to know where the information was obtained.”

In line with the reviewer’s recommendations, Table 1 has been substantially modified. The data are extracted from technical data sheets for packaging systems on the market, supplied by a producer of packaged fish fillets with whom we have collaborated (private data). To our surprise, the data sheets are sometimes marked as confidential (data not available).

Please find below the revised table (without the track changes mode in order to improve readability):

Vacuum packaging (VP)

Modified atmosphere packaging (MAP)

Flexible top film

polymer and main layer thicknesses

OPA*/PE**

(15µm/50 µm)

PE/PA***/EVOH****/PA/PE

(global film thickness 24 µm)

WVTR

less than 11 g/m2/day for 90% RH at 38°C, ASTM E96

18 g/m2/day

for 100% RH at 38°C, ASTM F1249

OTR

less than 40 cm3/(m2.day.bar)

for 0% RH at 23°C, ASTM D3985

24 cm3/(m2.day.bar) (permeance)

for 0% RH at 23°C, ASTM D3985

Semi-rigid tray

polymer and main layer thicknesses

PA/PE (80 µm/30 µm)

PET*****/PE (300 µm/30 µm)

WVTR

less than 5 g/m2/day for 90% RH at 38°C, ASTM F1249

data unavailable

OTR

less than 50-55 cm3/m2/day

for 75% RH at 23°C, ASTM D3985

less than 5 cm3/tray.day

(test conditions unavailable)

*OPA: Oriented PolyAmide, **PE: PolyEthylene, ***PA: PolyAmide, ****EVOH: Ethylene Vinyl Alcohol, *****PET: Poly-Ethylene Terephtalate

Many word extension of abbreviations are repeatable. For example TVB-N is mentioned in line 53 and 436. please check other abbreviations.

Thank you for this observation, this has now been carefully checked throughout the revised manuscript and changed accordingly.

Many paragraphs are short (<3 paragraphs), loss coherence to other paragraphs. It is suggested to combine to previous or next paragraphs.

This has been done in the revised manuscript in which several paragraphs were merged.

In the section of TVB-N and TMA (line 432), there are information related to limit of acceptability for various food products. It is recommended that the limit of accepatbility needs to overviewed in a table to know different limits for different fish products.

Thank you for this suggestion, a Table has been added in the revised manuscript (Table 2, page 11).

In Table 2, What is "molecule" mean here? VOCs?

Yes, you are right, the title of this column has been changed accordingly.

In Table 3, Author should shorten the information in the column of quality parameter. Take the important information

The information in the column of quality parameter of Table 3 has been shortened.

Reviewer 3 Report

1. Line 9: Please describe non-processed fish. Is it fresh and/ minimally processed fish?

2. Line 177-179 [Table 1]: What does OPA stand for? PE, PET?

3. Line 473: Does each fish species/type has different VOCs?

    Does the GCMS analysis able to differentiate the VOCs among fish species/type?

4. Line 513: Table 2 should be introduced in the text first.

     Dynamic headspace is the extraction method. Please add GC/MS or SIFT/MS to the extraction method, for example, Dynamic headspace/GC/MS.

     Please add the column information to the Analytical method.

5. Line 876-885: Both Appendix A and Appendix B should be deleted.

Author Response

Please see the attachment,  or preferably see Coverletter PDF file since attached MSword file was uploaded before reviewer  2 comments.

Round 2

Reviewer 1 Report

The changes reported in the previous review were accepted and the text is now better understood.